

# A neural machine translation method based on split graph convolutional self-attention encoding

Fei Wan[1] and Ping Li[2]

[1] School of Management, Hefei University of Technology, Hefei, Anhui, China
[2] School of Information Engineering, Fuyang Normal University, Fuyang, Anhui, China

## ABSTRACT

With the continuous advancement of deep learning technologies, neural machine translation (NMT) has emerged as a powerful tool for enhancing communication efficiency among the members of cross-language collaborative teams. Among the various available approaches, leveraging syntactic dependency relations to achieve enhanced translation performance has become a pivotal research direction. However, current studies often lack in-depth considerations of non-Euclidean spaces when exploring interword correlations and fail to effectively address the model complexity arising from dependency relation encoding. To address these issues, we propose a novel approach based on split graph convolutional self-attention encoding (SGSE), aiming to more comprehensively utilize syntactic dependency relationships while reducing model complexity. Specifically, we initially extract syntactic dependency relations from the source language and construct a syntax dependency graph in a non-Euclidean space. Subsequently, we devise split self-attention networks and syntactic semantic self-attention networks, integrating them into a unified model. Through experiments conducted on multiple standard datasets as well as datasets encompassing scenarios related to team collaboration and enterprise management, the proposed method significantly enhances the translation performance of the utilized model while effectively mitigating model complexity. This approach has the potential to effectively enhance communication among cross-language team members, thereby ameliorating collaborative efficiency.

## INTRODUCTION

In 2009, aiming to expand its global influence and underscore its commitment to customer security and transparency, the HSBC Bank introduced the slogan ''Assume Nothing.'' However, when translating this slogan into various other languages, an issue arose where a literal translation rendered it as ''Do Nothing,'' leading to adverse effects on its brand image. HSBC Bank had to invest $10 million to rectify this translation error. Translation issues not only impact corporate marketing efforts but can also significantly affect the corporate management processes of multinational companies. In numerous multinational

Corresponding author
Ping Li, apple151691@126.com

corporations, project management often involves teams comprising members from around the world. Challenges such as cross-border collaboration, multilingual communication, and cultural differences are inevitable. In international project environments, the translation of team communications becomes especially crucial, as it facilitates the transcendence of cultural barriers, enabling more effective communication and thereby enhancing overall project quality (*Pérez, 2002*; *Almashhadani & Almashhadani, 2023*). Given the complexity levels and scales of team projects, professional translation teams are often required to satisfy project demands. Machine translation, recognized as a tool that can reduce translation workloads and lower costs, is widely acknowledged as an effective approach (*Plaza-Lara, 2020*).

Machine translation aims to transform one natural language into another, facilitating cross-lingual information exchange (*Eria & Jayabalan, 2019*). Early approaches such as statistical machine translation (SMT) employed statistical models to establish translation rules and features for the translation process (*He, Liu & Lin, 2008*). With the advancement of deep learning techniques, neural machine translation (NMT) has swiftly emerged as a novel paradigm in the machine translation field (*Bahdanau, Cho & Bengio, 2015*). In contrast with traditional SMT methods, NMT employs neural network models to directly map source language sequences to target language sequences, bypassing intricate feature engineering and alignment issues. As a result, NMT exhibits enhanced flexibility and expression capabilities.

Despite these remarkable achievements, NMT still faces several challenges. Conventional NMT models typically rely solely on extensive bilingual corpora for training purposes, lacking a profound grasp of language structures and semantics. To achieve enhanced NMT performance, researchers in the field have begun incorporating linguistic knowledge, such as grammar rules (*Donaj & Sepesy, 2022*; *Peters et al., 2018*) and semantic information (*Su et al., 2021*; *Song et al., 2019*). Embedding such knowledge into models can assist them in better understanding and generating target language sentences. However, the introduction of linguistic knowledge often necessitates the construction of more intricate model architectures to delve into deep-seated linguistic insights. Additionally, extra data preprocessing and encoding steps are required for parallel corpora, substantially elevating the complexity and computational costs of the developed model.

In addition to integrating linguistic knowledge, scholars have further achieved enhanced translation performance by altering the structures of NMT models. For instance, *Miculicich et al. (2018)* introduced a hierarchical attention network (HAN) model to capture contextual information in a structured and dynamic manner. They integrated the acquired representation information into the original NMT architecture and demonstrated that utilizing the HAN model architecture in both the encoder and decoder enables the NMT model to complementarily extract additional contextual information from the context. Building upon the HAN model, *Maruf, Martins & Haffari (2019)* proposed a selective attention network (SAN) model that employs sparse attention to selectively focus on relevant sentences within the context of the given document. By combining hierarchical attention mechanisms based on sentence- and word-level context information, they obtained improved translation results on an English-German dataset. In contrast with

mainstream methods that often focus on applying network enhancements to encoders, (*Li et al., 2022*) employed a pretrained encoder combined with a bidirectional decoder. They introduced optimization strategies involving alignment-based code switching and dynamic dual masking, leading to significant machine translation performance improvements in autoregressive NMT tasks. These enhancements enabled the model to better capture the dependency relationships between the source and target languages and improved its capacity to model long-distance dependencies. However, for more intricate syntactic structures and semantic issues, the incorporation of deeper linguistic knowledge is often necessary to enhance the effectiveness of the utilized model.

In response to the aforementioned challenges, we propose an NMT approach based on split graph convolutional self-attention encoding SGSE. This method builds upon the standard transformer sentence translation model (*Vaswani et al., 2017*) with the following enhancements. First, it explores the syntactic dependency relations within the source corpus and constructs a syntax dependency graph. The words acquired from the source language are treated as nodes within this graph, and the extracted syntactic dependency relations serve as edges in the syntax dependency graph. Subsequently, by employing multiple rounds of message passing and aggregation, dependency graph convolution is employed to obtain graph convolutional semantic encodings. A self-attention network is constructed based on these graph convolutional semantic encodings, resulting in graph convolutional self-attention encoding. Finally, the transformer's self-attention network and the constructed graph convolutional self-attention network are split and concatenated based on their embedding dimensions, enabling encoding fusion. The fused encoding is utilized as the input of the encoder, thereby enhancing its contextual understanding and improving the resulting NMT performance.

The main contributions of this article are as follows.

(1) The study innovatively explores latent syntactic information in the corpus, utilizing it as additional knowledge to enhance the encoding accuracy of words in the corpus. Through this approach, we effectively improve the translation performance of neural machine translation models, enabling them to more accurately capture grammatical structures and generate more natural and precise translation results.

(2) Significant progress has been achieved in model enhancement. By adopting different network split coefficients, we successfully integrate graph convolutional self-attention encoding with the original context encoding. This innovative approach not only reduces the number of parameters in the fused NMT model but also effectively enhances the model's inference speed. This approach is crucial for improving the practicality and efficiency of the model, particularly in the context of team communication efficiency and communication quality in management activities, presenting significant potential for improvement.

## RELATED WORK

### Syntactic dependency relations

To enhance the translation capabilities of NMT models, many researchers have attempted to incorporate syntactic knowledge into such models. This syntactic knowledge includes

grammar rules, syntactic dependency relations, and semantic roles, providing the utilized model with additional insights into the structures and semantics of sentences. Embedding syntactic knowledge into NMT models enables better target language sentence comprehension and generation. One common approach involves combining syntactic knowledge as supplementary features or constraints with the mapping relationship between the source and target languages. For example, *Alqaisi (2023)* improved bilingual word embeddings by integrating syntactic dependency features into the NMT training process and demonstrated that pretrained NMT models with syntax dependencies achieved superior translation results. *Bugliarello & Okazaki (2019)* employed a self-attention mechanism to learn word embeddings and adjusted the weights of the source syntax self-attention in the encoder based on the syntactic distances between words, achieving translation results surpassing those of a transformer. *Pu & Sima'an (2022)* noted that the existing syntax-enhanced NMT models typically use a single most probable unlabeled parse or a set of best unlabeled parses derived from an external parser. They proposed concurrently transferring the parser's uncertainty and labeled syntactic knowledge to a transformer to achieve improved machine translation performance. Another approach involves introducing prior syntactic knowledge constraints during the model training process to guide the generation of target language sentences while adhering to grammar rules and syntactic relations. *Wu et al. (2018)* established a sequence dependency framework that considers both the source and target languages and utilized contextually constructed semantic dependencies derived from syntactic trees to attain enhanced translation performance. *Gong et al. (2022)* introduced a syntax-guided self-attention neural network using additional syntax awareness as bias information to guide and adjust the attention weights in the original transformer attention distribution. *Wan et al. (2023)* introduced syntactic dependency information to enhance the grammatical expression capacity of the model for the source language in NMT. They adopted a multilevel grammar evaluation method, exploring scalable translation approaches associated with syntactic knowledge. These methods effectively leverage syntactic knowledge to boost NMT capabilities, and they have demonstrated superiority in terms of handling complex syntactic structures and semantic issues.

## Graph neural networks

As approaches that integrate traditional syntactic knowledge, graph neural networks (GNNs) have been widely employed in recent years to enhance the translation capabilities of NMT models. GNNs constitute a category of deep learning techniques that are specifically designed to handle graph-structured data, proficiently capturing the relationships between nodes. In the NMT domain, source language sentences can be construed as graph structures, with each word serving as a node and with syntactic or dependency relationships constituting edges. The introduction of GNNs had enabled the modeling of the syntax relationships within source language sentences, integrating these relations into the model's representation.

For instance, *Marcheggiani, Titov & Bastings (2018)* constructed a semantic network based on predicate-argument structures, employing graph convolutional networks (GCNs) to update the word embeddings of the nodes within the semantic network.

Through this process, the model effectively uncovers the syntactic knowledge concealed within the semantic network. *Li et al. (2023)* proposed a dynamic graph convolutional translation architecture, utilizing a syntax graph structure as its input and generating a sequential output. In this framework, the decoder simultaneously handles source feature representations and their corresponding syntax graphs. By jointly modeling and generating target translations, this approach enhances the NMT model's grasp of syntactic knowledge and subsequently elevates the resulting translation quality. Additionally, *Nguyen et al. (2023)* encoded the Universal Conceptual Cognitive Annotation (UCCA) *via* a GNN for additional encoding purposes. They fused UCCA encodings into transformer word embeddings, effectively enhancing the translation capabilities of their NMT model. Through the incorporation of GNN-based feature extraction, the model attained an improved comprehension of the source language sentence's structure and semantics, leading to more accurate and coherent target language translation outcomes.

Integrating syntactic knowledge and GNN techniques aids models in gaining a deeper understanding of linguistic structures and semantic relationships, ultimately enhancing the final translation quality and fluency. However, the incorporation of supplementary syntactic knowledge and network structures could potentially augment the complexity of the model, resulting in increased computational resource demands and prolonged training and inference times.

In contrast with the existing works, we enhance the process of semantically extracting the source syntactic relations within the convolutional encoding of the dependency graph by additionally constructing a self-attention network. Simultaneously, the transformer's self-attention network and the constructed graph convolutional self-attention network are split and concatenated based on their embedding dimensions, enabling encoding fusion. This encoding fusion process not only reduces the complexity of the NMT model structure but also enhances the translation performance of the model.

## METHODS

To effectively enhance the efficiency and quality of team communications in management activities, this study explores an NMT approach based on SGSE. The overall model framework of SGSE is illustrated in Fig. 1.

The right-hand side of the figure presents the machine translation module of the model, while the left-hand side reveals the detailed steps required to obtain the graph convolutional encodings. To clearly illustrate the application of our proposed method, we deliberately select two sentences from a parallel corpus as examples to demonstrate the process of constructing dependency graphs between the source and target languages. This graph convolutional encoding approach aids in more effectively capturing the syntactic structures and contextual relationships between sentences in cross-language communication scenarios.

During the translation process, the SGSE model employs the same parallel encoding and decoding strategy as that of the standard transformer model, processing all the sentences derived from the input parallel corpus. For the purpose of analyzing syntactic

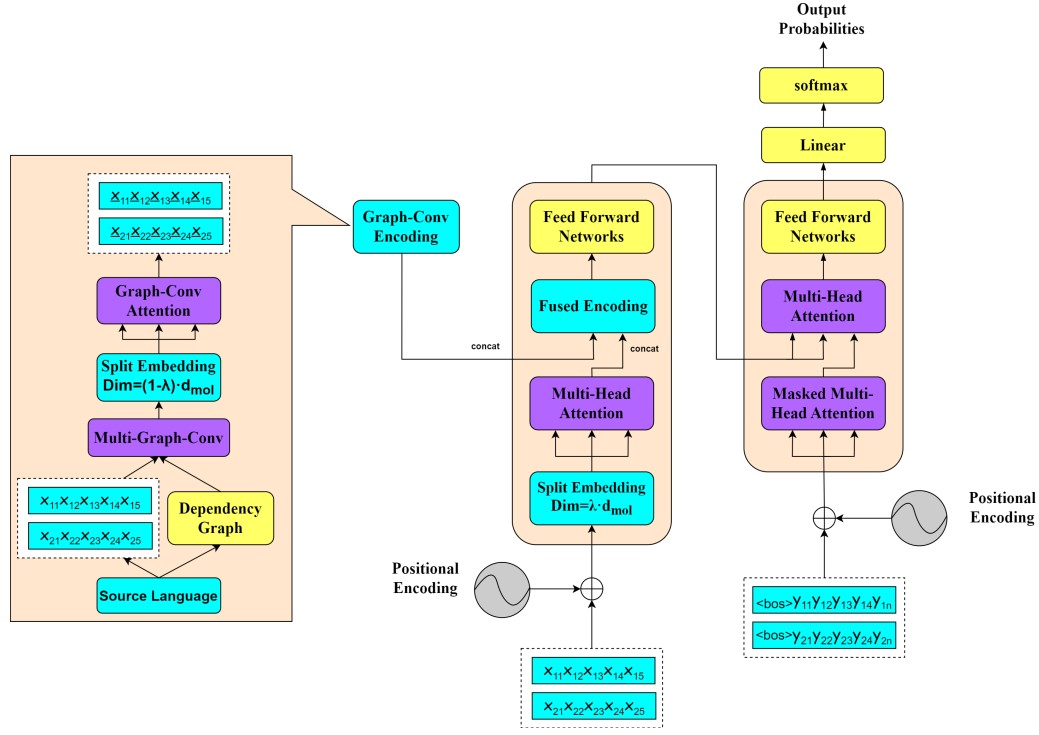

**Figure 1** NMT model framework based on SGSE.

relationship modeling, we construct a source dependency graph based on the source language corpus and utilize the word vectors generated from the source language corpus as the feature representations for the nodes in the source dependency graph. By applying multilayer graph convolutions, we obtain syntactic relationship-based graph convolutional semantic encodings (as detailed in Constructing Syntax Dependency Graphs). To acquire graph convolutional self-attention encodings for the current source input, we perform word embedding splitting operations that are complementary to the encoding-side splitting coefficients on the obtained graph convolutional semantic encodings, leading to the construction of a graph encoding self-attention network (as detailed in Syntactic Semantic Encoding Network). Before and after the self-attention computations on the model's encoding side, we conduct splitting and concatenation operations on the graph convolutional encodings, fusing the graph convolutional self-attention encodings (as detailed in Integration of Syntactic Semantic Encoding and Machine Translation). Through this fused encoding approach, the SGSE model substantially reduces the number of required parameters, enhances its computational efficiency, and simultaneously learns contextual information and syntactic knowledge from the source language corpus, resulting in improved machine translation performance.

## Constructing syntax dependency graphs

We utilize the syntactic dependency information derived from the source language corpus. By extracting these syntactic relationships, syntactic relation triplets are generated, and

based on these triplets, a syntax dependency graph is constructed. The generated syntax dependency graph provides richer and more accurate grammatical information for the subsequent NMT task, thereby aiding the model in better comprehending the structure and semantics of the source language sentence.

First, performing syntactic dependency analysis on the source language involves segmenting the source language into word-level units to extract the syntactic relationships between words. Assuming that $M$ sentences are contained in the source language, the segmentation of the m-th sentence $X_m$ in the source language is depicted as shown in Eq. (1).

$$X_m = x_{m1}, x_{m2}, x_{m3}, \ldots, x_{mN} \tag{1}$$

where $N$ represents the maximum number of words in the m-th sentence and $x$ corresponds to each word in the sentence.

Next, a pretrained syntax dependency analysis model is employed to perform dependency parsing on the source corpus. This model is trained on extensive corpora and is capable of automatically identifying the dependency relationships among words in sentences, such as subject-predicate relationships and verb-object relationships. Utilizing the syntactic dependency relationships, a set of syntactic relationship triplets is constructed for all M sentences in the source language, as depicted in Eq. (2).

$$\Gamma_M = \left\{ \left( x_{mn_1}, R_{\left( x_{mn_1}, x_{mn_2} \right)}, x_{mn_2} \right) \mid 0 < m \leq M, 0 < n_1 < n_2 \leq N \right\} \tag{2}$$

where $x_{mn_1}$ and $x_{mn_2}$ are two words in the m-th row of the source language that have a syntactic dependency relationship. These nodes correspond to the head node and tail node in the syntactic relationship triplet, respectively. $R_{\left( x_{mn_1}, x_{mn_2} \right)}$ represents the syntactic relationship between words $x_{mn_1}$ and $x_{mn_2}$, which is denoted as "$R$" in the subsequent descriptions in this article. The corresponding edge in the syntactic relationship triplet is not considered directional in our approach.

Finally, the obtained set of relationship triplets $\Gamma_M$ is used to construct a syntactic dependency graph $G$ based on the source language, as shown in Eq. (3).

$$G = \{ (V, E) \mid V \in \{x_{mn}\}, E \in \{R\}, 0 < m \leq M, 0 < n \leq N \}. \tag{3}$$

Here, $V$ represents the set of nodes in the syntactic dependency graph, and $E$ represents the set of edges. In the context of this article's approach, the influence of the strengths of the syntactic relationships on the weights of the edge relationships is not considered.

## Graph convolutional semantic encoding

For the m-th sentence in the source language, $X_m = x_{m1}, x_{m2}, x_{m3}, \ldots, x_{mN}$, the embedding layer of the transformer is employed to obtain word embeddings for each word in the sentence, as shown in Eq. (4).

$$E_m = E(x_{m1}, x_{m2}, x_{m3}, \ldots, x_{mN}) \tag{4}$$

where $E_m \in R^{N \times d_{mol}}$ and $d_{mol}$ represents the dimensionality of the word embeddings.

The approach presented in this article utilizes the word embeddings of each word as the feature encodings for the corresponding nodes in the constructed syntactic dependency graph. To achieve graph convolutional semantic encoding, a series of message passing and aggregation operations are performed on the syntactic dependency graph. Each node exchanges information with its adjacent nodes and updates based on the features of the surrounding nodes, gradually aggregating the contextual semantic information acquired from the neighboring nodes. The process of obtaining a graph convolutional semantic encoding for the m-th source sentence is illustrated in Eq. (5).

$$
E_m^{(l+1)} = \begin{cases} \sigma\left(D_{tgt}^{-\frac{1}{2}} A D_{adj}^{-\frac{1}{2}} E_m^{(l)} W^{(l)}\right) & l \geq 1 \\ E_m & l = 0 \end{cases}
\tag{5}
$$

where $D_{tgt}$ represents the input degree matrix of the target nodes, $D_{adj}$ represents the output degree matrix of the neighbor nodes, $A$ represents the adjacency matrix of the target nodes, and $E^{(l)}$ represents the encoding output of the l-th layer GCN. When $l$ is 0, the NMT model does not use the GCN and instead uses the initial word embedding representation $E_m$. $E_m^{(l)}$ represents the feedforward network layer added after the l-th layer of graph convolution, and $\sigma$ represents the process of message aggregation.

Notably, we do not employ an additional embedding encoding layer to represent the node features within the syntactic dependency graph. Instead, the original word embedding representations derived from the transformer are utilized. Consequently, in comparison with the traditional transformer model, the graph convolutional semantic encoding process introduced in this article does not require supplementary training parameters.

## Syntactic semantic encoding network

The method proposed in this article is built upon the transformer model introduced by *Vaswani et al. (2017)*, which has become the prevailing approach in the field of NMT. Additionally, the transformer model serves as a fundamental building block for language models such as ChatGPT. The transformer model departs from traditional NMT models that rely on methods such as recurrent neural networks (RNNs) (*Sutskever, Vinyals & Le, 2014*), convolutional computations (*Gehring et al., 2017*), and attention mechanisms (*Eriguchi, Hashimoto & Tsuruoka, 2016*). Instead, it fully embraces self-attention mechanisms to learn the contextual representations of words.

Building upon the foundation of the transformer model, we introduce a strategy to partition the initial word embedding representations. This partitioning process is subsequently used to tailor the input and output layers of the self-attention networks according to specific splitting ratios. For the m-th sentence, the split multihead self-attention computation is detailed in Eqs. (6) to (8). Formulas (6) and (7) are derived from *Vaswani et al. (2017)*.

$$
MHSA(Q_{ms}, K_{ms}, V_{ms}) = \left[head_1^s; \ldots; head_H^s\right] W_{ms}
\tag{6}
$$

$$
head_h^s = softmax\left(\frac{Q_{ms} K_{ms}^T}{\sqrt{d_k}}\right) V_{ms}
\tag{7}
$$

$$Q_{ms}, K_{ms}, V_{ms} = (\lambda \cdot E_m + PE)(\lambda \cdot W_Q, \lambda \cdot W_K, \lambda \cdot W_V) \quad 0 < \lambda < 1. \tag{8}$$

Here, the term MHSA denotes a multihead self-attention mechanism, wherein multiple parallel attention heads are employed. $head_h^s$ represents an individual self-attention head obtained after network pruning. $Q_{ms}$, $K_{ms}$, and $V_{ms}$ denote the query, key, and value vectors, respectively, obtained through the splitting of the embedding representations and the pruning of the self-attention network. $W_{ms}$ represents the parameter matrix to be learned by the model. $\lambda$ signifies the splitting coefficient of the embedding representation, which also serves as the pruning ratio for the self-attention network.

To more effectively extract features from the graph convolutional semantic encoding while simultaneously reducing parameter complexity and maintaining the NMT performance of the constructed model, we introduce an additional syntactic encoding self-attention network. This network takes the graph convolutional semantic encoding as its input. The computation process of the syntactic encoding self-attention network is illustrated by Eqs. (7) to (11).

$$MHSA(Q_{dep}, K_{dep}, V_{dep}) = \left[ head_1^{dep}; \ldots; head_H^{dep} \right] W_{dep} \tag{9}$$

$$head_h^{dep} = softmax\left( \frac{Q_{dep} K_{dep}^T}{\sqrt{d_k}} \right) V_{dep} \tag{10}$$

$$Q_{ms}, K_{ms}, V_{ms} = (1 - \lambda) E_m^{(l+1)} \left[ (1 - \lambda) \cdot W_Q, (1 - \lambda) \cdot W_K, (1 - \lambda) \cdot W_V \right] 0 < \lambda < 1. \tag{11}$$

Here, $head_h^{dep}$ represents an individual self-attention head in the syntax encoding self-attention network. $Q_{dep}$, $K_{dep}$, and $V_{dep}$ denote the query, key, and value vectors of the syntax encoding self-attention network, respectively. $W_{dep}$ represents the parameter matrix that the model learns.

In contrast with Eq. (8), we do not introduce positional encoding during the graph convolutional semantic encoding process. This design considers the characteristics of GCNs: their node connections are constructed based on syntactic dependency relations, and GCNs are not sensitive to the relative positional relationships between nodes. Given that GCNs inherently account for the relative positional relationships between nodes when handling nonsequential-structured data, reintroducing positional encoding would result in redundant encoding steps and potentially impact the performance of the model.

### Integration of syntactic semantic encoding and machine translation

Building upon the foundation of graph convolutional semantic encoding, this study explores the fusion of this encoding process with split self-attention network encoding. In 'Syntactic Dependency Relations', we first employ the split self-attention network to segment the initial word embedding representation, resulting in split self-attention network encodings that retain contextual semantic information. Subsequently, in 'Graph

Neural Networks', we obtain graph convolutional syntactic semantic encodings, which encompass semantic information obtained by constructing independent syntactic encoding self-attention networks.

During the fusion process, we seamlessly match the dimensions of the two encodings with the initial word embedding dimensions through the settings of the splitting coefficients in Eqs. (8) and (11). By concatenating these two encodings in Eq. (12), we successfully combine the contextual semantic information with the syntactic dependencies among different words in the source language.

$$E_{fu} = \left[ MHSA(Q_{ms}, K_{ms}, V_{ms}); MHSA\left(Q_{dep}, K_{dep}, V_{dep}\right) \right]. \tag{12}$$

Here, $E_{fu}$ represents the fused word encoding, and $[\,;\,]$ denotes the concatenation operation.

Through this fusion approach, we maintain consistent encoding dimensions, avoiding unnecessary information losses or issues arising from dimension mismatches. The fused encodings retain the observed contextual information while also encompassing richer syntactic relationships, enhancing the model's understanding of the structure and semantics of the source sentence. Consequently, this method achieves significant improvements in machine translation tasks.

Compared to the traditional transformer model, the SGSE model greatly reduces the number of required parameters. This advantage stems primarily from the optimized design of the split self-attention network and the syntactic semantic self-attention network. During the self-attention computation, the parameter count is determined mainly by the query (Q), key (K), and value (V) dimensions, where the number of parameters needed for each dimension equals the square of the word embedding dimensionality. For a single encoder, the parameter reduction achieved by the SGSE model compared to the traditional transformer model is illustrated in Eq. (13).

$$\begin{aligned} P_{reduce} &= P_{reduce}(Q, K, V) + P_{reduce}(MHSA) \\ &= 3\left\{ d_{mol} - (\lambda d_{mol})^2 - [(1-\lambda) d_{mol}]^2 \right\} + \left[ H d_{mol} - H(\lambda d_{mol})^2 - H[(1-\lambda) d_{mol}] \right]^2. \\ &= 2(3+H)\lambda(1-\lambda) d_{mol}^2 \end{aligned} \tag{13}$$

Here, $P_{reduce}(Q, K, V)$ represents the reduction in the parameter count of the "query, key, and value" parameter matrices in the self-attention mechanism, while $P_{reduce}(MHSA)$ represents the reduction in the parameter count of the linear transformation matrices in the multihead self-attention mechanism. $H$ stands for the number of attention heads, $d_{mol}$ represents the dimensionality of the word embeddings, and $\lambda$ denotes the splitting coefficient.

As inferred from Eq. (13), it is evident that the parameter reduction achieved by our approach is linearly proportional to the number of self-attention heads and quadratically proportional to the word embedding dimensionality. This signifies that under equivalent model dimensions, our proposed method can efficiently utilize the available computational resources, substantially diminishing the parameter count of the constructed model. Consequently, this leads to decreased computational costs in terms of training and inference.

**Table 1** Software and hardware details for the simulations.

| Experimental parameters | Detailed Information |
|---|---|
| CPU | Intel(R) Core(TM) i7-10700K |
| Memory | 64GB |
| Hard Disk | 6TB |
| GPU | NVIDIA GeForce RTX 4070 Ti 12GB |
| Operating System | Microsoft Windows 10 |
| Programming Language | Python 3.9.17 |
| Toolkit | torch 1.13.1+cu117 |
| | fairseq 0.12.2 |
| | dgl 1.1.1+cu117 |
| | hanlp 2.1.0 |

**Table 2** Statistics of the parallel sentence pairs in the different datasets.

| Dataset | IWSLT14de-en | IWSLT20es-de | WMT14fr-en | IWSLT17zh-en |
|---|---|---|---|---|
| train | 160250 | 260871 | 160538 | 204954 |
| val | 7284 | 11857 | 8484 | 3300 |
| test | 6750 | 13613 | 3003 | 5566 |

# RESULTS AND DISCUSSION

## Experimental settings

To ensure the comparability of the simulation results, we conducted simulation experiments in the same software and hardware environment. The details of the software and hardware resources, as well as crucial toolkit information, are presented in Table 1.

To validate the effectiveness of the proposed approach, we selected parallel corpora from the International Workshop on Spoken Language Translation (IWSLT) and Workshop on Machine Translation (WMT) conferences as experimental datasets, IWSLT is an international conference focused on spoken language translation, while WMT is an international conference dedicated to machine translation. The source corpora for the experiments encompass four language pairs: German-English (de-en), Spanish-German (es-de), French-English (fr-en), and Chinese-English (zh-en). For the de-en translation task, the data were sourced from the TED speech data in the IWSLT 2014 corpus (https://wit3.fbk.eu/2014-01). The es-de translation task utilized the TED speech data from IWSLT 2020 (https://wit3.fbk.eu/2020-01), while the fr-en translation task employed the News-Commentary dataset from the WMT 2014 conference (https://www.statmt.org/wmt14). The zh-en translation task relied on the TED speech data contained in the IWSLT 2017 corpus (https://wit3.fbk.eu/2017-01-c). Detailed information about the sample sizes of the parallel corpora for each dataset is provided in Table 2.

All parallel corpora underwent subword segmentation through byte pair encoding (BPE) (*Sennrich, Haddow & Birch, 2016*). To constrain the memory usage of the model, while learning the BPE encoding rules, we set a uniform upper limit of 10,000 BPE tokens.

Additionally, the parallel corpora underwent preprocessing steps, including normalization, tokenization, lowercase conversion, and corpus cleaning.

Building upon the transformer framework, this study extended its capabilities and conducted comparisons with several baseline methods. These baseline methods include the standard transformer sentence translation model (*Vaswani et al., 2017*) based on parallel corpora; Mix-models (*Shen et al., 2019*), which employs ensemble learning to capture translation diversity through multimodel blending; LightConv (*Wu et al., 2019*) and Linformer (*Wang et al., 2020*), both of which aim to achieve model parameter and computational complexity reductions; and the NMT model uses dependency parse graphs and graph attention networks (DP-GAT), which utilizes syntactic relationships to enhance its translation effect. The original DP-GAT model (AMR+GAT) was proposed by *Nguyen, Pham & Dinh (2020)*, while the baseline model employs a GAT to encode semantic information derived from dependency parse graphs.

In this study, we implemented the proposed NMT model based on SGSE encoding using the open-source Fairseq toolkit (*Ott et al., 2019*). We utilized the HanLP toolkit to extract the dependency syntactic relationships between words in the source language and employed the open-source DGLtoolkit (*Wang, 2019*) to perform message passing and aggregation operations on the dependency graph. Regarding the selection of the baseline models, both the proposed approach and all baseline methods adopted the transformer_base model parameters. The encoder and decoder consisted of six layers each, with eight attention heads per layer and a word embedding dimensionality of 512. During the model training process, the learning rate was set to $10^{-4}$, and the employed loss function was label_smoothed_cross_entropy. To ensure acceptable training efficiency and memory usage, each sentence's maximum length was capped at max-tokens of 4,096. Given concerns about excessive smoothing due to an excessive number of graph convolutional layers, the number of graph convolutional layers was set to 3, which was consistent with the literature (*Chen et al., 2020*). All the experimental models were trained for 150 epochs on an RTX 4070Ti GPU. The detailed information regarding the experimental parameters is presented in Table 3.

During the model evaluation stage, a beam search algorithm was employed with a beam size of 5. To accurately gauge the translation quality of each model, we employed the "multi-bleu.Perl" script from the Moses toolkit (*Koehn et al., 2007*) to calculate BLEU scores for NMT tasks, providing an objective assessment of the models' translation performance.

## Main experimental results

This section aims to validate the effectiveness of the proposed method in terms of its machine translation performance while also assessing the simplicity of its model. Table 4 provides a summary of the BLEU scores produced by both the baseline NMT models and the proposed method across four translation task test sets. Figure 2 illustrates the loss value

**Table 3   Parameter settings for model training.**

| NMT model | Parameter | Value |
|---|---|---|
| Transformer | / | Same as SGSE |
| Mix-models | num-experts | 3 |
| LightConv | / | Same as SGSE |
| Linformer | / | Same as SGSE |
| DP-GAT | / | Same as SGSE |
| SGSE | Encoder layers | 6 |
| | Decoder layers | 6 |
| | Attention heads | 8 |
| | Embed dimension | 512 |
| | Learning rate | 10-4 |
| | Max tokens | 4096 |
| | Training epochs | 150 |
| | Graph layers | 3 |
| | Optimizer | Adam |
| | Dropout | 0.3 |
| | Weight decay | 5e−5 |
| | Activation function | ReLU |
| | Lr scheduler | inverse-sqrt |
| | Wormup | 4000 |

**Table 4   Comparison among the BLEU scores of different models on public datasets.**

| | NMT-Model | IWSLT14de-en | IWSLT20es-de | WMT14fr-en | IWSLT17zh-en | |
|---|---|---|---|---|---|---|
| | | | | | BLEU score | Parameters |
| Baseline | Transformer | 33.49 | 23.03 | 24.24 | 19.67 | 73.9 Million |
| | Mix-models | 33.06 | 23.60 | 24.21 | 19.80 | 74.2 Million |
| | LightConv | 33.60 | 23.25 | 25.69 | 20.02 | 55.8 Million |
| | Linformer | 29.71 | 20.80 | 20.57 | 16.58 | 58.4 Million |
| | DP-GAT | 34.37 | 24.28 | 27.11 | 20.85 | 74.9 Million |
| Our Method | SGSE | 34.62 | 23.97 | 26.94 | 20.92 | 69.2 Million |

and BLEU score variations exhibited on the validation set during the training process of each model.

Compared to the standard sentence translation model (the transformer), our approach makes full use of the syntactic dependency information within the source language, leading to a significant translation quality enhancement. Across the IWSLT14 de-en, IWSLT20 es-de, WMT14 fr-en, and IWSLT17 zh-en datasets, our method achieves BLEU score improvements of 1.13, 0.94, 1.70, and 1.25, respectively, over the transformer. Notably, in the zh-en translation task, our method also reduces the number of required model parameters by 6.4%.

In contrast with Mix-models, which employs ensemble learning techniques, our approach not only demonstrates superior translation capabilities but also obviates the

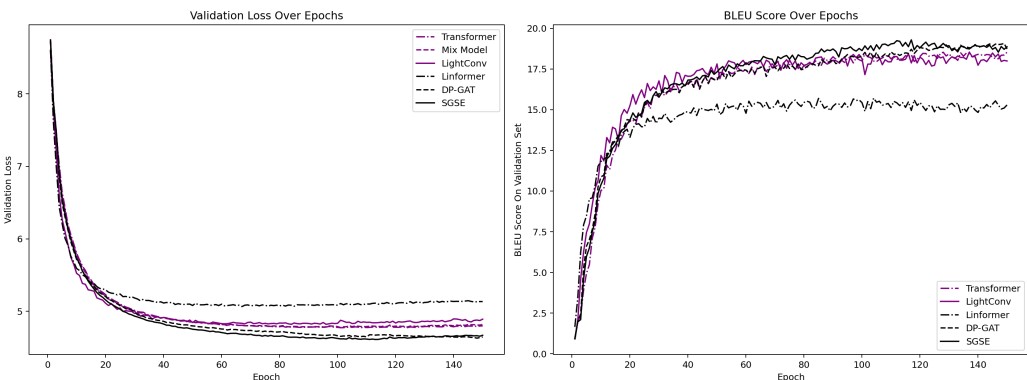

**Figure 2** **Variations in the loss and BLEU values produced on the validation set.**

need to train multiple independent NMT models, resulting in a notable reduction in model training time. When juxtaposed with the LightConv and Linformer models, which also utilize structural optimizations, our approach exhibits clear advantages. Compared to LightConv, our method achieves BLEU score improvements of 1.02, 0.72, 1.25, and 0.80.

Moreover, in comparison with the DP-GAT model, which utilizes syntactic dependency relations and employs an additional GAT to extract syntactic information, our approach attains leading BLEU scores in the IWSLT14 de-en and IWSLT17 zh-en tasks and performs comparably to the DP-GAT model in the remaining two translation tasks. Additionally, our method achieves a 7.7% reduction in the number of required model parameters.

## Impacts of different splitting coefficients

This section aims to investigate the impacts of different splitting coefficients, denoted as "$\lambda$", for the proposed method on the translation performance of the NMT model. In the experiments of this study, we perform the splitting operation on the word embeddings and graph convolutional encodings of the transformer to adjust the relative contributions of the self-attention computation and graph convolutional encoding during the encoding process. The splitting coefficient is used to specify the proportions of the vector dimensions used in self-attention computation and graph convolutional encoding processes. It is divided into eight equal parts to ensure that the dimensionality of self-attention is divisible by the number of attention heads, maintaining computational efficiency. The experimental results are presented in Table 5.

The experimental results demonstrate that the splitting coefficient significantly impacts the model's performance and parameter count. When the splitting coefficient is small, the influence of the original transformer embeddings on the final fused encoding is minimal, and the graph convolutional encoding process dominates. In this case, the model's performance gradually improves as the splitting coefficient increases. Conversely, when the splitting coefficient is large, the original transformer embeddings start to play a dominant role, and the impact of graph convolutional encoding diminishes. In this scenario, the contribution of the contextual semantics becomes more prominent. When the splitting coefficient is set to 0.75, the model requires fewer parameters (approximately

**Table 5  Experimental comparison among the results obtained in Chinese-English translation tasks with different splitting coefficients.**

| split_rate λ | Parameters | BLEU produced on val set | BLEU produced ontest set |
|---|---|---|---|
| 0 | 73.9 million | 1.24 | 2.29 |
| 0.125 | 71.1 million | 17.73 | 19.04 |
| 0.25 | 69.1 million | 18.19 | 19.75 |
| 0.375 | 67.9 million | 18.33 | 19.89 |
| 0.5 | 67.5 million | 18.04 | 19.38 |
| 0.625 | 67.9 million | 18.84 | 20.19 |
| 0.75 | 69.1 million | 19.29 | 20.92 |
| 0.875 | 71.1 million | 19.07 | 20.65 |
| 1 | 73.9 million | 18.53 | 19.67 |

**Table 6  BLEU scores produced by translation models on parallel sentence pairs with different sentence lengths.**

| Sentence length | Number of sentences | Transformer | SGSE | BLEU Increase |
|---|---|---|---|---|
| less than 10 | 1,102 | 24.05 | 25.34 | 1.29 |
| 10∼14 | 1,136 | 21.10 | 22.46 | 1.36 |
| 15∼20 | 1,119 | 20.54 | 22.12 | **1.58** |
| 21∼32 | 1,200 | 19.92 | 20.66 | 0.74 |
| over 32 | 1,009 | 18.05 | 19.46 | 1.41 |
| All sentences | 5,566 | 19.67 | 20.92 | 1.25 |

69.1 million) and achieves the highest BLEU score (20.92). This suggests that at this particular splitting coefficient, the model strikes a favorable balance between performance and parameter efficiency.

Notably, when the splitting coefficient is 0, the model utilizes only dependency graph convolution embeddings as feature inputs, whereas when the splitting coefficient is 1, the model relies solely on contextual semantic embeddings as feature inputs. Clearly, contextual semantic embeddings serve as the primary source of information during the model's translation process, and the introduction of dependency graph convolution embeddings enriches information from the perspective of grammatical structure, thereby enhancing the translation performance of the model.

## Model performance comparison across different corpus lengths

In this section, we conduct a comprehensive comparative analysis of the performance of our proposed method and the transformer model across various sentence lengths. We aim to showcase their performance variations within different length ranges. To achieve this, we partition the test set into five groups based on the lengths of English sentences. This division enables us to meticulously investigate the performance of both methods under different length conditions. The experimental results are presented in Table 6.

We conduct experimental comparisons on parallel corpora from different groups. In the group with sentence lengths ranging from 15 to 20, a BLEU score improvement of 1.58 is observed. In the other groups, the BLEU score improvements fall between 0.7

**Table 7  Experimental comparison results obtained on team communication datasets.**

| Parallel corpora | Number of parallel pairs | Transformer | | DP-GAT | | SGSE | |
|---|---|---|---|---|---|---|---|
| | | BLEU score | Inference Time(s) | BLEU score | Inference Time(s) | BLEU score | Inference Time(s) |
| NLP-CEPARACFIN | 100 | 15.46 | 1.5 | 15.91 | 2.2 | 15.75 | 1.8 |
| TED-Enterprise | 1,173 | 22.41 | 16.5 | 24.09 | 18.4 | 24.05 | 15.4 |
| TED-Management | 1,241 | 22.07 | 17.4 | 23.81 | **19.5** | 23.68 | **16.7** |
| UN-Conference | 4,000 | 8.53 | 55.9 | 9.55 | 61.4 | 9.72 | 50.4 |
| NC-Enterprise | 5,969 | 10.60 | 83.3 | 11.35 | 91.1 | 11.49 | 73.2 |
| NC-Management | 3,270 | 10.85 | 45.7 | 11.86 | 50.3 | 11.71 | 43.0 |

and 1.41. We posit that longer sentences tend to encapsulate more intricate internal syntactic relationships, thereby furnishing the model with a wealth of valuable syntactic information, which aids in optimizing its translation performance. Conversely, shorter sentence lengths might lead to an internal syntactic information reduction, limiting further model enhancement. Notably, owing to the inherent characteristics of machine translation, excessively long sentences might encounter constraints during decoding due to factors such as beam search algorithms, thus restricting the ability to obtain substantial performance gains.

## Performance analysis of the model on the team communication dataset

In this section, we shift our focus to real-world application scenarios such as team collaboration, business operations, and management. We conduct an in-depth comparison between the SGSE model and several baseline models in terms of translation accuracy and speed.We utilize a carefully curated group of datasets to comprehensively evaluate the performance of the proposed model. Among them, the NLP-CEPARACFIN dataset (https://magichub.com/) is an open-source parallel corpus with financial domain activities in Chinese and English. The UN-Conference dataset (https://conferences.unite.un.org/ UNCorpus) originates from United Nations conferences and serves to test the diversity and challenges of corpora. Furthermore, we meticulously select enterprise-related parallel corpora from the TED2020 and News Commentary17 datasets, which are designated as TED-Enterprise and NC-Enterprise, respectively. Additionally, we extract parallel corpora associated with business and government management, labeled TED-Management and NC-Management, respectively. All the experimental results are presented in Table 7.

Compared to the transformer model, the SGSE model demonstrates superior translation performance across six distinct datasets. Particularly, in scenarios involving large-scale samples, the SGSE model exhibits faster translation speeds than the transformer. Additionally, it is worth noting that the SGSE model achieves translation performance on par with that of the DP-GAT method while maintaining a significant advantage over the DP-GAT model in terms of translation speed.

Furthermore, the SGSE model is more effective in extracting syntactic structure information present in the corpus. For instance, in the 1,336th line of the TED-Management dataset, the reference translation is ''I joined Unilever in 1976 as a management trainee in India.'' The translation generated by the SGSE model is ''In 1976, I joined Unilever as a management training student in India.'' In contrast, the Transformer's translation is ''In 1976, I joined Unilever in India as a management training training.'' Clearly, the translation produced by the Transformer exhibits serious grammatical errors, while the SGSE model maintains a sentence structure consistent with the reference translation, with differences only in the translation of specialized terms such as ''management trainee.''

These findings collectively reveal that the SGSE model can significantly enhance the quality and efficiency of collaboration within multilingual teams. This advantage presents the possibility of optimizing the cross-cultural communication and management paradigms within multinational enterprises, with the potential to foster closer cooperation among different language groups within an organization.

## CONCLUSIONS

We introduce a novel NMT approach named SGSE, which incorporates split graph convolutional self-attention encoding into the constructed model. By effectively harnessing syntactic dependency relationships and optimizing the design of self-attention networks, our approach achieves a translation performance improvement while significantly reducing the number of required model parameters. Through experiments conducted on various standard conference datasets, we showcase the outstanding performance of the SGSE model in multilingual translation tasks. Furthermore, we validate the practicality and efficiency of our method in real-world team collaboration and management scenarios.

In future research, we will strive to further optimize our method to better cater to the demands of practical applications. Specifically, we plan to pursue the following two directions. First, we plan to conduct in-depth validation of our proposed methods on a broader range of datasets, encompassing bidirectional translation datasets with complex syntactic structures and deep semantics, such as German, Tamil, and others. This will contribute to a more comprehensive assessment of the applicability and universality of our approach. Second, in the face of scarce parallel corpora, we will explore leveraging the constructed syntactic dependency graph to predict the potential relationships between nodes, aiming to further enhance the translation quality achieved under small-sample conditions such as team management scenarios.

### Funding

This research was funded by the Key Research Project of Natural Science in Colleges and Universities of Anhui Province, China (Nos. KJ2021A1253), and the Startup fund for doctoral scientific research, Fuyang Normal University, China (No. 2022KYQD0010). The

funders had no role in study design, data collection and analysis, decision to publish, or preparation of the manuscript.

## Grant Disclosures

The following grant information was disclosed by the authors:

The Key Research Project of Natural Science in Colleges and Universities of Anhui Province, China: Nos. KJ2021A1253.

The Startup fund for doctoral scientific research, Fuyang Normal University, China: No. 2022KYQD0010.

## Competing Interests

The authors declare there are no competing interests.

## Author Contributions

- Fei Wan performed the experiments, analyzed the data, performed the computation work, prepared figures and/or tables, authored or reviewed drafts of the article, and approved the final draft.
- Ping Li conceived and designed the experiments, analyzed the data, authored or reviewed drafts of the article, and approved the final draft.

## Data Availability

The data is available at Zenodo: Fei Wan. (2024). Split-Attention-dataset. https://doi.org/10.5281/zenodo.10483878.

The code is available at GitHub and Zenodo:

- https://github.com/HenryVanHuy/Split-Attention.
- HenryVanHuy. (2024). HenryVanHuy/Split-Attention: alpha (v1.0.0). Zenodo. https://doi.org/10.5281/zenodo.10511660.

The Third-Party data is available at:

- IWSLT 14 de-en: https://wit3.fbk.eu/2014-01
- IWSLT20 es-de: https://wit3.fbk.eu/2020-01
- WMT14 fr-en: https://www.statmt.org/wmt14/translation-task.html#download
- IWSLT17 zh-en: https://wit3.fbk.eu/2017-01-c
- NLP-CEPARACFIN: https://magichub.com/
- TED: https://opus.nlpl.eu/TED2020/zh&en/v1/TED2020
- UN-Conference: https://conferences.unite.un.org/UNCorpus
- NC: https://data.statmt.org/news-commentary/v17/.

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
