# Peer review of "A neural machine translation method based on split graph convolutional self-attention encoding"

_PeerJ Computer Science, doi:10.7717/peerj-cs.1886_

## Round 0.1 · original submission · Minor Revisions

Dear authors,

Thank you for submitting your work. Reviewers have now commented on your article and suggest major revisions. When submitting the revised version of your article, it will be better to address the following suggestions clearly.

1- Are the simulation results taken from the equal conditions? There is not any discussion. Add further details on how simulations were conducted. Similarly, resource and system characteristics could be added to Tables for clarity. Add further details on how simulations were conducted. The paper lacks the running environment, including hardware and software details. Please write the software and hardware details for the simulations. The analysis and configurations of experiments should be presented in detail for reproducibility. It is convenient for other researchers to redo your experiments and this makes your work easy acceptance. A table with parameter settings for experimental results and analysis should be included in order to clearly describe them.

2- Some equations should be used with correct citations. They seem as if they are proposed and used firstly in this paper. This approach cannot be acceptable.

3- Many of the equations are part of the related sentences. Attention is needed for correct sentence formation.

4- Some mathematical notations are not rigorous enough to correctly understand the contents of the paper. The authors are requested to recheck all the definition of variables and further clarify these equations. Definitions of all variables and their intervals should be given.

5- All of the values for the parameters of all algorithms selected for comparison should be given.

Best wishes,

·

Basic reporting

The authors communicated the methods, research problem, and their findings in unambiguous and professional English. The descriptions of tables and figures are appropriate and will be easily understood by a person with a medium or advanced level of English skills. Their mathematical expressions and the descriptions of each variable in each equation adhere to academic standards. However, there are some places where a little effort is required to understand the notation.

The paper covers enough literature to understand the background of the study, and it is concise in explaining and inferring its objectives. However, more explanation towards various datasets and their backgrounds would have been beneficial. Readers might need to check the recent dataset or have prior knowledge about it.

The article's structure is well-organized, and references to tables and figures are appropriate, providing a seamless reading experience. There are no redundant references, and every reference to the table or diagram in the entire article is appropriate.

Regarding the number of papers cited, everything is necessary to explain the work, although they could have referred to more publications for their dataset references. Apart from this, everything was on point.

The description of variables, equations, and terms is well-defined, making the reading of the paper comfortable. The sequencing is clear without any back-and-forth references. However, the author didn't explain the abbreviation "MHSA" used in equations 13, 12, 9, and 6. Additionally, when it comes to the dataset, although researchers in this field are expected to know the abbreviations "IWSLT" and "WMT," it would have been helpful if they were explained upon first usage.

Experimental design

The research falls within the aim and scope of the journal. I strongly recommend publishing this paper in the journal because it delves into the core of NMT and NLP architecture. However, it would have been even more impactful if the researchers had validated their claims in various areas within the same domain.

The research question posed is, "Does the existing architecture used widely today possess knowledge of the semantic aspects of the source language?" This is a significant study and finding, especially considering that complex languages like Chinese (zh) and German have intricate grammar and semantic structures compared to languages like English and French. The problem mentioned is prevalent in today's scenario. However, the paper does not present direct evidence of how the semantic knowledge problem is solved, except for the improvement of the results. The authors could have illustrated their proposal better by depicting a few handpicked sentences that performed poorly, not incorporating semantic structure with the existing method, and comparing them with the new method. Including a few direct sentence-to-sentence examples would have enhanced the clarity of the paper's proposal.

The paper employs the standard metric BLEU to compare results. Although based on text string matching, BLEU is widely used in the research area. The proposed method and provided code are well-written and explained. Regarding the Python code, some vital classes like buildHomo and buildHete lack documentation, but this can be included later and shouldn't impact the publication.

Validity of the findings

The impact of the findings is assessed in a well-known dataset, namely WMT and IWSLT. However, results are only provided for a very limited number of language pairs. When experimenting with the semantic structure of a language, it would have been great if the authors had included languages like German, Tamil, and other semantically rich languages. I particularly appreciate the experimentation with the Chinese. The concluded result is well aligned with the research question and the proposed method.

Additional comments

Overall, I enjoyed the research question, proposed method, and conclusion. In future research, please include more datasets and try to compare the results of language pairs in both ways (e.g., en-es and es-en).

·

Basic reporting

This paper proposes split graph convolutional self-attention encoding(SGSE), a novel approach for neural machine translation (NMT) that constructs non-Euclidean syntax dependency graphs and integrates split self-attention networks.

From lines 100 - 102: “Subsequently, by employing multiple rounds of message passing and aggregation, dependency graph convolution is employed to obtain graph convolutional semantic encodings…”. How many rounds? Does the number of rounds of message passing and aggregation affect the dependency graph convolution and overall performance?
From line 108: “The main contributions of this article are as follows…”. The main contribution section could be simplified into one paragraph with the answers to two questions: “What’s the novelty” and “What’s the improvement”. The discussion of the experiment results can be expanded in the Experimental Design section.
From lines 189 - 191: “This encoding fusion process not only reduces the complexity of the NMT model structure but also enhances the translation performance of the model.” Would appreciate more information on how the complexity is reduced. For example, compared to the GNN-based feature extraction, why attention network constructed from the dependency graph is superior?
Thanks for providing a clear list of datasets and GitHub links.
The paper maintains a clear structure and provides a comprehensive list of literature studies on neural machine translation models.

Experimental design

From lines 457 - 459: “When the splitting coefficient is set to 0.75, the model requires fewer parameters (approximately 69.1 million) and achieves the highest BLEU score (20.92). This suggests that at this particular splitting coefficient, the model strikes a favorable balance between performance and parameter efficiency.” abd Table 3: “Experimental Comparison Among the Results Obtained in Chinese-English Translation Tasks with Different Splitting Coefficients”. It can be seen that the split_rate chosen near 0.75 are 0.625 and 0.875. What are the motivations for choosing such numbers? If 0.75 split_rate produces fewer parameters, what about rates such as 0.70 or 0.8?
From line 495: “Particularly, in scenarios involving large-scale samples, the SGSE model exhibits faster translation speeds than the transformer.” What is the exact number of comparisons between the two models and what is the exact model architecture(number of layers, embedding dimension, …) More specifically, is there any specific improvement regarding the inference performance of the SGSE model across all datasets(especially large-scale ones)?

Validity of the findings

The findings of the paper are valid as it is supported by experiments conducted on open datasets. For a more accurate representation of the performance of the proposed model, please address comments in the Experimental Design section.

Additional comments

Thanks for the opportunity to review the manuscript. Please address the comment above.

·

Basic reporting

1) The authors proposed an NMT method based on convolutional self-attention.
2) Authors should align texts in paragraphs as ‘justify.’

Experimental design

1) the authors should add an ablation study in the experiment chapter.

Validity of the findings

1) The difference in BLEU Scores between the baselines and the proposed method is insignificant. They should work on that to improve further.

Additional comments

1) The paper is well-written and well-structured.
2) Overall, the work is novel and satisfactory.

---

## Round 0.2 · accepted · Accept

Dear authors,

Thank you for the revision and for clearly addressing all the reviewers' comments. I confirm that the paper is improved and addresses the concerns of the reviewers. In light of this revision, your paper is now acceptable for publication.

Best wishes,

·

Basic reporting

Thanks for submitting the rebuttal and polishing the paper. The authors provided more background on various sections of the paper, especially regarding the dataset and experiment design. The drawn conclusion aligns with the proposed research topic and has the potential for more discussions in various domains.

Experimental design

The author has addressed the questions regarding the splitting ratios and provided explanations regarding the model performance, especially regarding the inference speed comparison in models.

Validity of the findings

Overall, the paper proposed a novel solution in the NLP domain and is well-written with sound datasets and experiments. The findings in the paper can be considered valid.

Additional comments

Thanks for submitting the revised paper.

·

Basic reporting

The paper is well-written with the necessary data.

Experimental design

improved in the revised version.

Validity of the findings

improved in the revised version.